# Mechanism of ATP hydrolysis dependent rotation of bacterial ATP synthase

Atsuki Nakano[1], Jun-ichi Kishikawa [1,2], Kaoru Mitsuoka[3] & Ken Yokoyama [1] ✉

$F_1$ domain of ATP synthase is a rotary ATPase complex in which rotation of central γ-subunit proceeds in 120° steps against a surrounding $\alpha_3\beta_3$ fueled by ATP hydrolysis. How the ATP hydrolysis reactions occurring in three catalytic αβ dimers are coupled to mechanical rotation is a key outstanding question. Here we describe catalytic intermediates of the $F_1$ domain in $F_oF_1$ synthase from *Bacillus PS3* sp. during ATP mediated rotation captured using cryo-EM. The structures reveal that three catalytic events and the first 80° rotation occur simultaneously in $F_1$ domain when nucleotides are bound at all the three catalytic αβ dimers. The remaining 40° rotation of the complete 120° step is driven by completion of ATP hydrolysis at $\alpha_D\beta_D$, and proceeds through three sub-steps (83°, 91°, 101°, and 120°) with three associated conformational intermediates. All sub-steps except for one between 91° and 101° associated with phosphate release, occur independently of the chemical cycle, suggesting that the 40° rotation is largely driven by release of intramolecular strain accumulated by the 80° rotation. Together with our previous results, these findings provide the molecular basis of ATP driven rotation of ATP synthases.

The majority of ATP, the energy currency of life, is synthesized by oxidative phosphorylation or photophosphorylation, catalyzed by ATP synthase[1–3]. F-type ATP synthases, $F_oF_1$, exist in the inner membrane of mitochondria, the plasma membrane of bacteria, and the thylakoid membrane of chloroplasts, while Archaea and some bacteria express the homologous V-type ATP synthases, called V/A-ATPases[4–8]. $F_oF_1$ consists of a hydrophilic $F_1$ domain containing three catalytic sites[9], and a hydrophobic $F_o$ domain housing a proton translocation channel[10,11] (Fig. 1a). The movement of protons through the translocation channel of $F_o$ drives rotation of the γ-stalk and this catalyzes the conversion of ADP to ATP at the $\alpha_3\beta_3$ dimer catalytic sites referred to as $\alpha_E\beta_E$, $\alpha_T\beta_T$, and $\alpha_D\beta_D$. Upon dissociation from the $F_o$ domain, the $F_1$ domain can catalyze the reverse reaction, i.e. hydrolysis of ATP driven by rotation of the central γ subunit inside the cylinder comprising $\alpha_3\beta_3$ (Fig. 1b, c). The catalytic sites are located at interface between the α and β subunit in each catalytic dimer. Most of the residues involved in ATP binding and hydrolysis are found in the β subunit, although residues in the α subunit are also involved[12,13].

The binding change mechanism of ATP synthesis at the $F_1$ domain by rotation of the γ subunit relative to $\alpha_3\beta_3$ was firstly proposed by P. Boyer[9,14]. According to this mechanism, the three catalytic dimers adopt different conformational states, and they interconvert sequentially between three the different conformations as catalysis proceeds. Strong support for this asymmetrical $F_1$ arrangement was confirmed by the X-ray crystal structure of mitochondrial $F_1$ ($MF_1$)[13], which revealed the three catalytic β subunits in different conformational states and with different nucleotide occupancy at the catalytic sites; closed $\beta_{DP}$ with ADP, closed $\beta_{TP}$ with ATP analog, and open $\beta_E$ with no bound nucleotide.

Rotation of $F_1$ driven by ATP hydrolysis was directly demonstrated in single molecule observations using bacterial $F_1$ from *Geobacillus stearothermophilus* (*Bacillus PS3* sp.)[15]. When using 40 nm gold beads with almost negligible viscous resistance, $F_1$ pauses at the 80° dwell position during the 120° rotation step at an ATP concentration close to the $K_m$[16]. Histogram analysis of the frequency of dwell times for each dwell position suggests a model in which binding of ATP to $F_1$ at 0°

[1]Department of Molecular Biosciences, Kyoto Sangyo University, Kamigamo-Motoyama, Kita-ku, Kyoto 603-8555, Japan. [2]Institute for Protein Research, Osaka University, 3-2 Yamadaoka, Suita, Osaka 565-0871, Japan. [3]Research Center for Ultra-High Voltage Electron Microscopy, Osaka University, Osaka, Japan. ✉ e-mail: yokoken@cc.kyoto-su.ac.jp

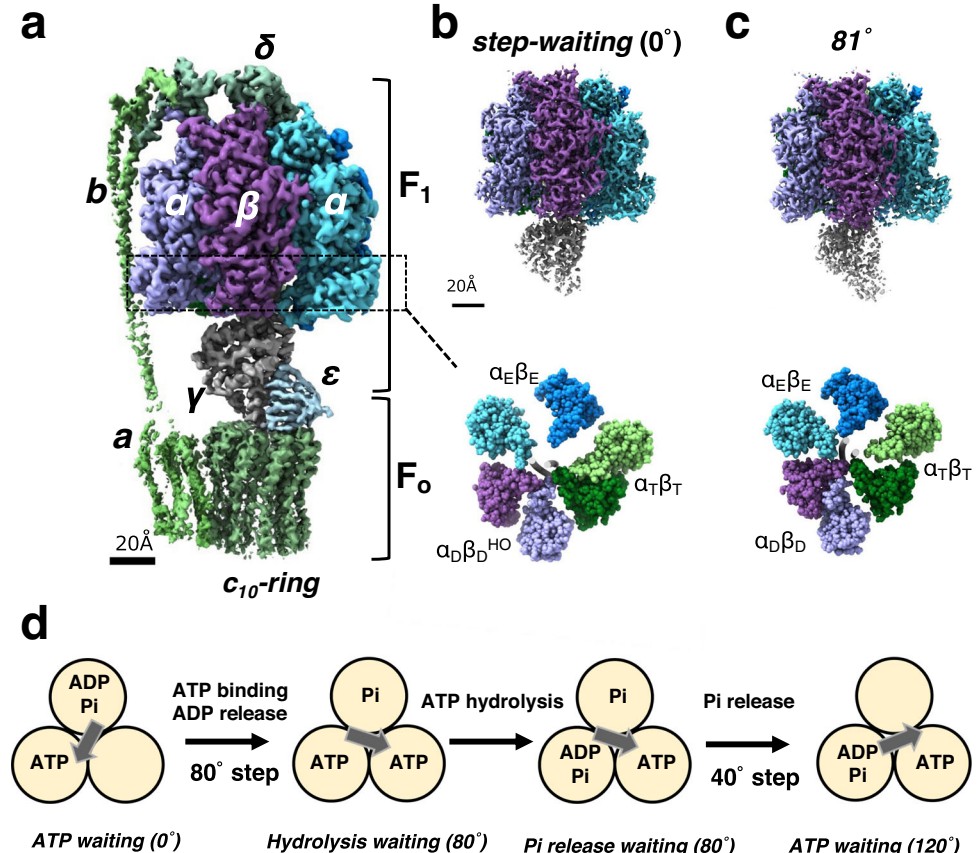

**Fig. 1 | Structures and rotation mechanism of $F_oF_1$. a** Cryo-EM structure of $F_oF_1$ ATP synthase in the *step-waiting* conformation. Each subunit is colored differently. The $F_1$ domain contains three catalytic αβ dimers which surround the γ subunit. **b**, **c** Side views (*upper*) and bottom section views (*lower*) of the $F_1$ domain in the *step-waiting* (0°) and 81° structure, respectively. The three catalytic dimers are represented different colors: $α_E$ (marine blue) and $β_E$ (light blue), $α_T$ (moss green) and $β_T$ (light green), and $α_D$ (purple) and $β_D$ (light purple). The $β_D$ subunit in *step-waiting* adopts a more open structure, termed as DHO. The γ subunits are represented as a gray tube in the center of both $α_3β_3$ sub-complexes. **d** A proposed scheme for chemo-mechanical coupling during a 120° rotation step of the $F_1$ domain driven by ATP. In this model, ATP binding to the $F_1$ domain immediately initiates the 80° rotation with an associated release of ADP. ATP is hydrolyzed at the 80° dwell position with no associated rotation of γ. It is the release of Pi from the enzyme which is suggested to drive the final 40° rotation.

causes an initial 80° rotation step of γ subunit, followed by a 40° rotation of the γ subunit due to hydrolysis of ATP and release of phosphate at the 80° dwell position (Fig. 1d)[16,17]. In addition, measurement of the rotational velocity of the rotating probe with viscous resistance or external force suggests that $F_1$ converts the hydrolysis energy of ATP into torque with high efficiency[18,19]. However, these single-molecule experiments only allow observation of the motion of the γ subunit and hence do not provide information on the changes occurring at each catalytic site.

In order to elucidate the entire $F_1$ rotation mechanism driven by ATP, we set out to determine the cryo-EM structures of the catalytic intermediates of the $F_1$ domain during rotation. We trap these intermediate states by freezing cryo-EM grids at different time points or under different reaction conditions allowing us to build up a picture of the chemo-mechanical cycle of these motor proteins step by step[20]. Previously, we determined several intermediate structures of V/A-ATPase from *Thermus thermophilus* revealing to a high level of detail the ATP hydrolysis cycle in these enzymes. The results showed that catalytic events occur simultaneously at the three catalytic sites of $A_3B_3$ stator, driving the 120° rotation of the central rotor[21]. Sobti et al. also performed snapshot analysis using a mutant $F_1$ exhibiting slow ATP hydrolysis and obtained structures corresponding to both the 0° and 80° rotation angles of γ subunit[22]. They proposed, however, a different rotation model of rotary ATPase, in which ATP binding and ATP hydrolysis drive the initial 80° rotation and the 40° rotation of $F_1$, respectively. As a result, the coupling of the chemo-mechanical cycle

of rotary ATPases remains controversial, 60 years after Boyer first predicted the rotary catalytic mechanism of ATP synthase.

Here, we describe multiple reaction intermediates of the wild type of $F_oF_1$ from *G. stearothermophilus* captured using cryoEM. These structural intermediates of the $F_1$ domain fill in the missing pieces in the rotation scheme and have allowed us to redraw the complete chemo-mechanical coupling scheme of rotation in $F_oF_1$ powered by ATP hydrolysis.

## Results
### Sample preparation and analysis

The ATP hydrolytic activity of *G. stearothermophilus* $F_oF_1$ is very low due to the up-form of the C-terminus of ε subunit[23,24]. In this study, we used the $ΔεCT$-$F_oF_1$ mutant, less susceptible to ε inhibition by truncation of ε-C-terminal helix than the wild-type[24]. The $ΔεCT$-$F_oF_1$ expressed in *E. coli* membranes was solubilized in DDM, then purified as described in Methods section. The $ΔεCT$-$F_oF_1$ exhibited ATPase activity, which obeyed simple Michaelis–Menten kinetics with a $V_{max}$ of 164 s$^{-1}$ and a $K_m$ of 30 μM (Supplementary Fig. 1a). The purified $ΔεCT$-$F_oF_1$ was subjected to nucleotide depletion treatment by dialysis against EDTA-phosphate buffer, in order to deplete endogenous nucleotide. The enzymatic properties of the nucleotide depleted enzyme (ND-$ΔεCT$-$F_oF_1$) were almost identical to the non-depleted enzyme (Supplementary Fig. 1b). In contrast to the results obtained by Paik et al. using $F_1$-ATPase[25], our purified $F_oF_1$ showed no initial lag phase (Supplementary Fig. 1c). Instead, the ATPase activity of the enzyme gradually

increased, indicating that $F_oF_1$ transitions from an initial inactive state to an active state without adopting the ADP inhibited state.

Both depleted and non-depleted forms of the purified enzymes were concentrated then subjected to a cryo-grid preparation as described below. The $\Delta\varepsilon CT$-$F_oF_1$ is referred to simply as $F_oF_1$, hereafter.

To exclude the possibility that the nucleotides present in the resulting structure were endogenous, we used $ND$-$F_oF_1$ without endogenous nucleotides for structural analysis at high [ATP]. When conducting reactions at low [ATP], the reaction solution typically contains 9 μM $F_oF_1$ and 25 μM ATP. Therefore, using $ND$-$F_oF_1$ at low [ATP] is likely to lead to ATP depletion in the solution, as most of the ATP will bind to the three α subunits in $F_1$ domain. To avoid this issue, we used $F_oF_1$ without nucleotide depletion treatment for structural analysis at low [ATP].

## Structures of 81° and *post-hyd* (83°) at high [ATP]

Cryo-grids were prepared using a reaction mixture containing 26 mM ATP and 9 μM $ND$-$F_oF_1$ to final concentration. The reaction mixture was incubated for 20 s at 25 °C, and then loaded onto a holey grid, followed by flash freezing. To prevent depletion of ATP in the reaction mixture due to the ATPase activity of $F_oF_1$, the concentration of ATP in the reaction mixture was set to 26 mM. This ensured that even after a 20-s reaction time, a saturating concentration of ATP remained in the solution, allowing us to obtain the structure under ATP-saturated conditions.

The flow charts showing image acquisition and reconstitution of the 3D structure of $F_oF_1$ at high [ATP] are summarized in Supplementary Fig. 2b. The particles containing three rotational states (154 k particles) were subjected to 3D classification focused on the $F_1$ domain in order to obtain sub-states of the $F_1$ domain. As the result of this classification, three sub-classes of the $F_1$ domain were obtained with different γ positions, termed the 80°, 100° and 120° structures (Supplementary Fig. 2b). Each structure was further classified by a masked classification.

The two $F_1$ domain 80° structures were obtained from 56 k particles using an $\alpha_D\beta_D$ covering mask (Supplementary Fig. 2b). The two 80° structures shared the canonical asymmetric hexamer which adopts the closed $\alpha_T\beta_T$, closed $\alpha_D\beta_D$, and open $\alpha_E\beta_E$ conformation (Fig. 2a, Supplementary Fig. 4). The angle of the γ subunits in the two 80° structures relative to the 0° structure were 81° and 83°, respectively (Figs. 2a, 3b). The nucleotide bound to the $\alpha_D\beta_D$ differed between the two structures. In the $\alpha_D\beta_D$ of the 81° structure, ATP was identified at the catalytic site (Fig. 2c, *left*) and we termed the structure as *81°*. In contrast, the 83° structure obtained at high [ATP] contained ADP and Pi and is designated post-hydrolyzed (*post-hyd*) (Fig. 2c, *center*). This indicates that ATP bound to $\alpha_D\beta_D$ is hydrolyzed at 81° or between 81° and 83°, and that the structural change due to ATP hydrolysis is small.

The overall structures of $\alpha_T\beta_T$ and $\alpha_E\beta_E$ in the 81° and *post-hyd* states are mostly identical, but the $\alpha_D\beta_D$ structures differ slightly. In the *post-hyd*, the *CT* domain of $\beta_D$ was in a slightly more open conformation relative to that in 81° (Fig. 2b).

In both 81° and *post-hyd*, ATP and Pi were identified at the catalytic sites of $\alpha_E\beta_E$. The Pi is occluded by β/164 K, β/190E, β/191 K, β/252D, β/256 R, and α/365R (Fig. 2d, Supplementary Fig. 6f). The Pi binding site structure of $\alpha_E\beta_E$ is highly similar to that of the yeast $F_1$ $\beta_E$ Pi-binding site[26]. Since $\beta_E$ has an open structure, the site occupied by Pi is separated from the site where ATP is bound. Therefore, in this state the γ-phosphate of ATP does not directly interact with the bound Pi. For other structures obtained under high [ATP], as described in the next section, ATP was identified at the catalytic sites of $\alpha_E\beta_E$ (Supplementary Fig. 4d). This indicates that $\alpha_E\beta_E$ is capable of binding of ATP to the catalytic site, regardless of the rotary angle of the γ subunit.

## Structures at 91°, 101° and 120° rotation angles at high [ATP]

Two additional structures found to be rotated a further 8° and 18° relative to the *post-hyd* (83°), respectively, were classified (Supplementary Fig. 2b). We termed these the 91° and 101° structures

(Fig. 3b). In both structures, ATP was identifiable at the catalytic site of $\alpha_E\beta_E$, but Pi is additionally bound to the 91° as well as the 81° and *post-hyd* forms (Fig. 2d and Supplementary Fig. 4d). In contrast, Pi is not present at the catalytic site of $\alpha_E\beta_E$ in the 101° structure, indicating that Pi is released during the 10° rotation of the γ subunit from 91° to 101°. The $\alpha_E\beta_E$ in 101° adopts a more open structure than that seen in the 91° structure due to movement of the *CT* domain of $\alpha_E$ (Fig. 3f and Supplementary Fig. 4d). This more open arrangement results in a more exposed catalytic site in $\alpha_E\beta_E$ and thus a decrease in affinity for Pi in the 101°. The geometry of the amino acid residues in $\beta_E$ that coordinate Pi changes due to the release of Pi (Fig. 3h), but no significant change in the conformation of the main chain in $\beta_E$ was observed (Fig. 3f), suggesting that the release of Pi directly does not cause the opening of $\alpha_E\beta_E$.

Two 120° structures were classified (Supplementary Fig. 2b) according to structural differences in $\alpha_T\beta_T$. One 120° structure, termed *step-waiting*, has a more closed $\alpha_T\beta_T$ than the second, termed 120°. $\alpha_T\beta_T$ in 120° is almost identical to that of the 101° structure (Fig. 3a and Supplementary Fig. 7a). The difference in $\alpha_T\beta_T$ between 120° and *step-waiting* is the result of movement of the *CT* domain of the $\alpha_T$ subunit (Supplementary Fig. 7c). Thus, sequential structural changes occur; a structural change from 101° to the 120° associated with the 19° rotation of the γ subunit, followed by closing of $\alpha_T\beta_T$ which occurs between 120° and *step-waiting*. Notably, the 19° step of the γ-subunit from the 101° to the 120° occurs independently of the catalytic cycle of ATP, such as Pi release and ATP hydrolysis in the $F_1$ domain.

Comparing the four conformational states, 81°, post-hyd, 91°, 101°, the difference in $\beta_D$ among these structures is more marked than that of $\beta_E$ and $\beta_T$. As the rotational angle of the γ subunit increases, the $\beta_D$ adopts a more open structure due to movement of the *CT* domain but remains bound to both ADP and Pi (Fig. 3c and Supplementary Fig. 5a, b). The motion of the *CT* domain in $\beta_D$ continues until the rotation angle of the γ subunit reaches 101° (Fig. 3a, c). The $\alpha_D\beta_D$ dimer in the 101° and 120° structures can be described as half open ($\alpha_D\beta_D^{HO}$). This conformational change of $\alpha_D\beta_D$ also occurs largely independently of the catalytic cycle of ATP, with the exception of the release of Pi occurring between 91° and 101°. This suggests that the opening of $\alpha_D\beta_D$ is largely driven by the release of strain on the molecule accumulated in the initial 80° rotation step.

## Structures of $F_1$ domain at low [ATP]

To obtain the structure of the $F_1$ domain awaiting ATP binding, $F_oF_1$ containing endogenous nucleotides was mixed with the reaction solution containing 25 μM ATP and 0.2 mg/ml of pyruvate kinase and 4 mM phosphoenolpyruvate in order to regenerate consumed ATP. The solution was incubated at 25 °C for 60 s and then loaded onto a holey grid followed by flash freezing. The flow charts showing image acquisition and reconstitution of the 3D structure of $F_oF_1$ at low [ATP] are shown in Supplementary Fig. 3b. From the combined 75 k particles for the 80° structure, three intermediate structures, two 83° and one 81° were classified. The two *post-hyd* structures are almost identical, including the angle of the γ subunit, apart from a slightly more open structure of the $\beta_D$ being evident in one of the forms.

Both 81° and *post-hyd* structures obtained at low [ATP] were very similar to the corresponding structures obtained at high [ATP] (Supplementary Table 1), but there were significant differences. In the 81° and *post-hyd* structures obtained at low [ATP], Pi was identified at the catalytic site of $\alpha_E\beta_E$, but no ATP density was observed (Fig. 4a, Supplementary Fig. 8a). In addition, ATP molecule was not found in other obtained structures at low [ATP] (Fig. 4b and Supplementary Fig. 8a) indicating that ATP does not bind to $\alpha_E\beta_E$ at low [ATP]. This contrasts with the binding of ATP to all structures of $\alpha_E\beta_E$ obtained at high [ATP] (Supplementary Fig. 4d).

The density of nucleotides in the $\alpha_D\beta_D$ at 81° under low [ATP] is not well defined (Fig. 4a, *right*). This implies that ATP and (ADP + Pi) are

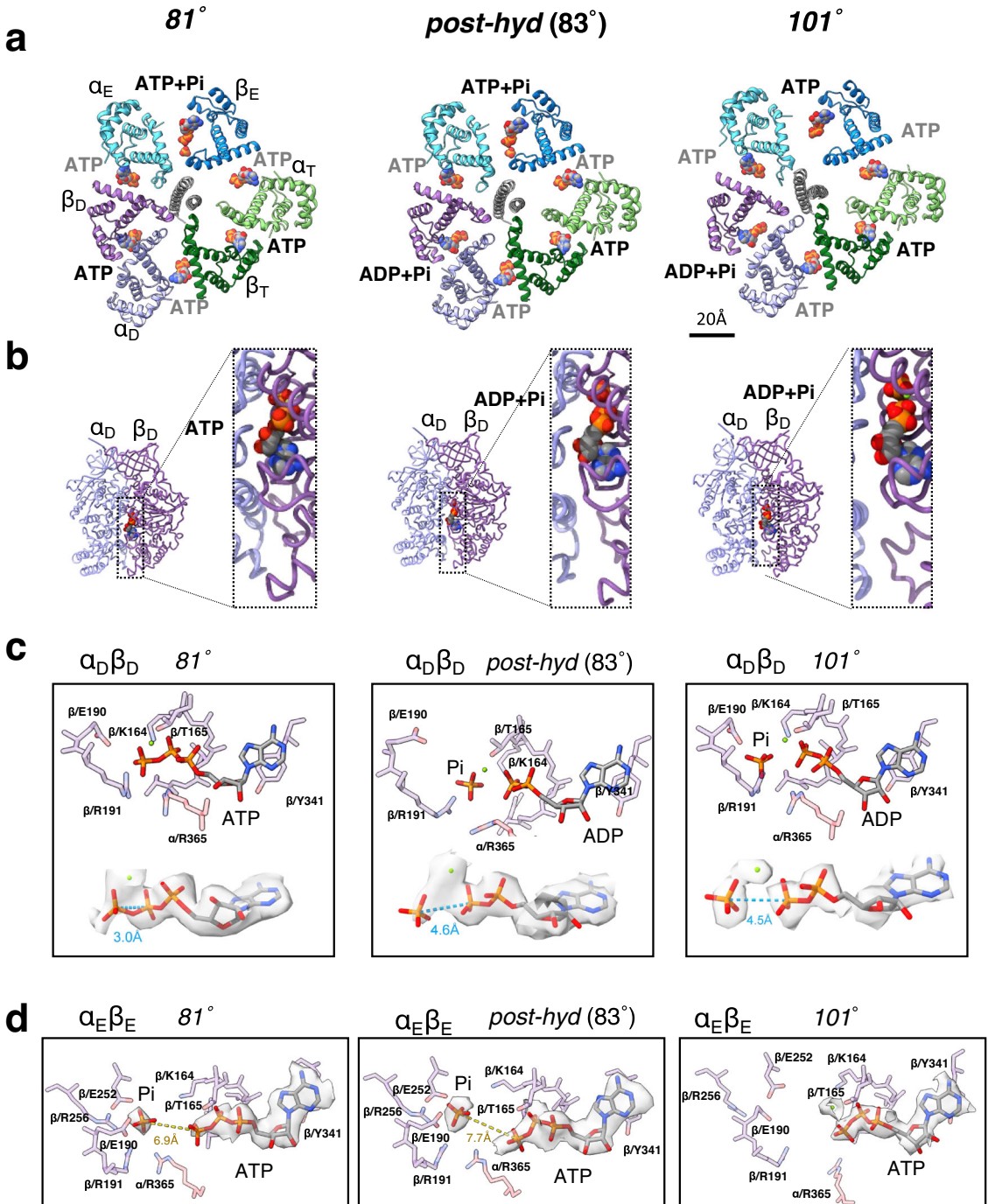

**Fig. 2 | Structure of 81° (left), *post-hyd* (center), and 101°(right) at high [ATP].**
**a** Cross section of the $F_1$ domain showing the catalytic sites viewed from the $F_o$ side. Each catalytic dimer is shown in ribbon representation and colored as detailed in Fig. 1. The bound nucleotides are represented as spheres. **b** Side view structure of $\alpha_D\beta_D$ dimer. The left panel is a magnified view of the catalytic interface at each $\alpha_D\beta_D$. **c** Structure of the catalytic site in $\alpha_D\beta_D$. Amino acid residues and bound nucleotide and Pi are represented as sticks. *Lower panels;* Stick representation model of ATP/ADP and Pi with EM density in $\alpha_D\beta_D$. The distance between γ and β phosphate of the ATP in each conformational state is shown in blue (Å). Each distance was calculated using Chimera software. **d** Structure of the catalytic site in $\alpha_E\beta_E$. The EM density of ATP / Pi is superimposed onto the model. The distance between Pi and γ phosphate of ATP is shown in yellow (Å).

in equilibrium at the catalytic site of $\alpha_D\beta_D$ in 81°. Additionally, this observation suggests that ATP hydrolysis at the catalytic site does not directly cause rotation of the γ subunit, consistent with the binding change mechanism where ATP synthesis at the catalytic site is independent of rotation of the γ subunit[1].

From the combined 332 k particles for 120° structure at low [ATP], three structures of the $F_1$ domain, 91° and 120° and *step-waiting*, were identified (Supplementary Fig. 3b). The 101° structure was not captured in this condition. At high [ATP], the $F_1$ domain of 120° and *step-*

*waiting* were composed of the $\alpha_D\beta_D^{HO}$ with ADP and Pi bound, $\alpha_T\beta_T$ with ATP, and $\alpha_E\beta_E$ with ATP (Supplementary Fig. 4c–e). However, at low [ATP], ATP nor Pi density was identified at the catalytic site of $\alpha_E\beta_E$ in either the 120° and *step-waiting* states. We refer to the *step-waiting* containing an empty $\alpha_E\beta_E$ as *ATP-waiting*.

In summary, ATP was not identified at $\alpha_E\beta_E$ in all five structures, 81°, post-hyd, 91°, 120°, *and ATP-waiting*, obtained at low [ATP]. Comparing these results with the nucleotide occupancy of $\alpha_E\beta_E$ in all structures obtained at high [ATP] indicates that ATP is capable of

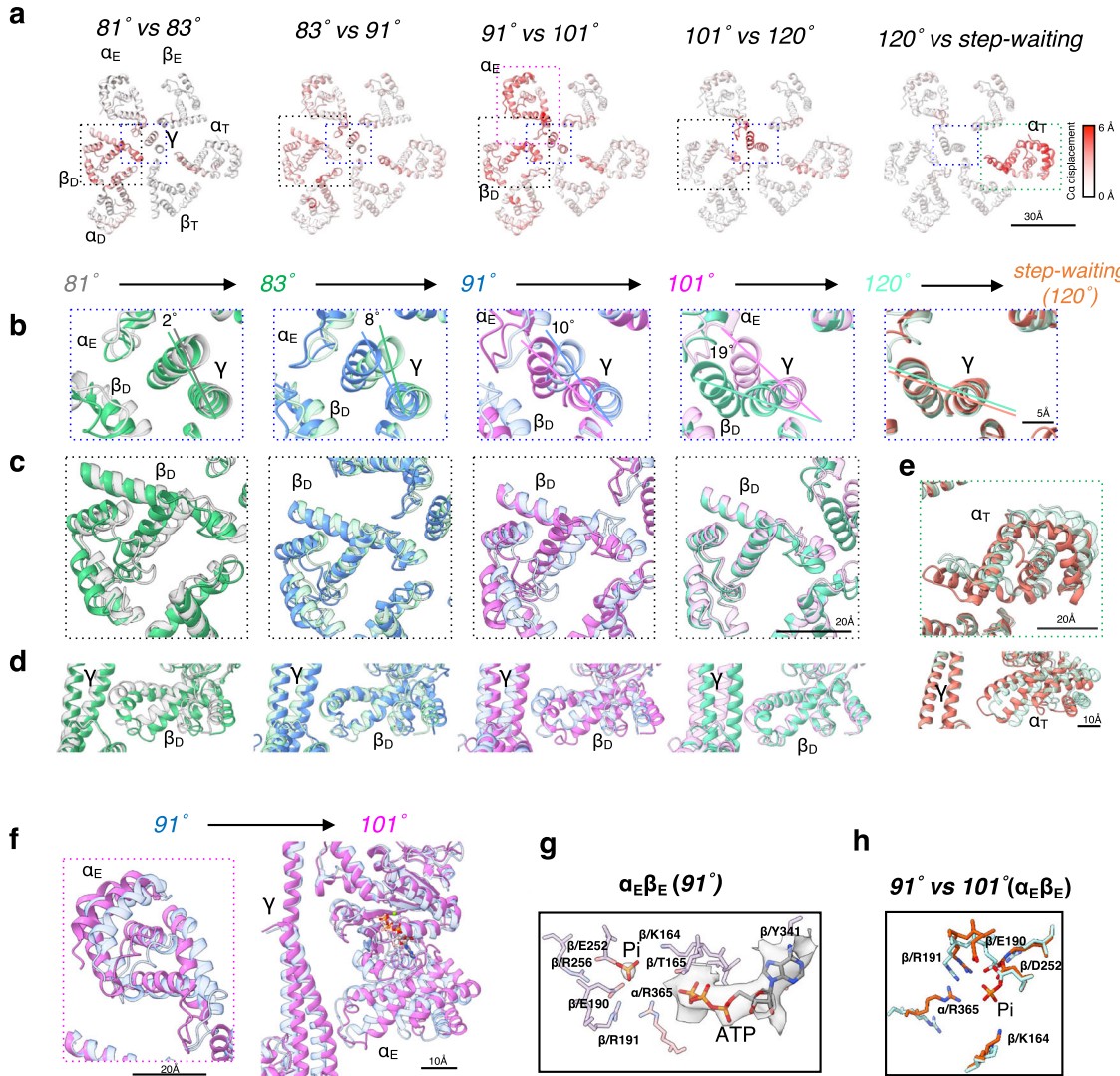

**Fig. 3 | Structure comparison of 6 intermediates captured during the 40° step at high [ATP]. a** Cross section of the F₁ domain showing the catalytic site. 81°, post-hyd, 91°, 101°, and 120° structures are arranged from left to right. The Cα displacement relative to the next structure (*right side*) is indicated by the red-white color gradient. The dashed square indicates the area of this figure shown in the zoomed in view in **b**–**f**. **b**–**f** Comparison of each structure to the following one. **b** compares the γ subunit and its surroundings, and the different rotation angles between each γ subunit and that of the next structure are indicated by different colored lines. **c** shows an upper view of β_D, **d** compares β_D from a side view, and **e**, **f** compare α_T (120° and step waiting) and α_E (91° and 101°), respectively. Each cartoon chain is colored as 81°(gray), 83°(green), 91°(blue), 101°(pink), 120°(light green), and *step-waiting* (orange). **g** Structures of the catalytic site of 91° in α_Eβ_E. The key side chains involved in coordinating Pi and nucleotide are shown in stick representation. Semi-transparent electron density for the Pi and nucleotide is shown over the stick representations. **h** Superimposition of the side chains of α_Eβ_E in 91°(orange) with those of α_Eβ_E in 101° (*light blue*).

binding to α_Eβ_E independent of the rotation angle of γ in the F₁ domain (Fig. 5).

## Rotation scheme of F₀F₁

In this study, we have captured multiple structures of F₁ domain and from these we can reveal the rotation scheme that occurs during the full 120° rotation (Fig. 5a–j, Supplementary movie 1). The first 80° rotation occurring via *step-waiting* of rotational state 1 is coupled with structural transition of three catalytic dimers, α_Eβ_E to α_Tβ_T, α_Tβ_T to α_Dβ_D, and α_Dβ_D^HO to α_Eβ_E. The resultant 81° is in rotational state 2 (Fig. 5d). We further classified the three rotational states of F₀F₁ from the single-particle images of each F₁ domain structure, enabling us to reproduce the 360° rotation scheme of F₀F₁ upon hydrolysis of the three ATPs (Fig. 5k). Comparison and interpretation of these intermediate structures provides us with several important insights into the coupling of the chemo-mechanical cycle of ATP hydrolysis driven rotation of F₀F₁.

## Tri-site mechanism

ATP binds to empty α_Eβ_E of the F₁ domain at any rotation angle, as shown in Fig. 5. Based on single molecule observation experiments for F₁-ATPase, the coupling scheme for chemo-mechanical cycle of F₁-ATPase proposed that rotation occurs simultaneously with ATP binding to the structure at a rotation angle of 0° (Fig. 1d)[16–18]. In contrast, structural snapshots of V-type ATP synthase (V/A-ATPase) revealed that the 120° rotations occur after all three catalytic sites are occupied by nucleotides (Supplementary Fig. 10a)[21]. The snapshot analysis of F₀F₁ structures in this study indicates that binding of ATP to empty α_Eβ_E in *ATP-waiting* results in *step-waiting* in which all three catalytic sites are occupied by nucleotides, then the first 80° rotation starts (Fig. 5c). This means that ATP binding to the F₁ domain does not immediately cause the 80° rotation. Our results clearly demonstrate that F₀F₁ and V/A-ATPase share a common tri-site mechanism in which nucleotide occupancy transitions between 2 and 3 binding sites during continuous catalysis[2].

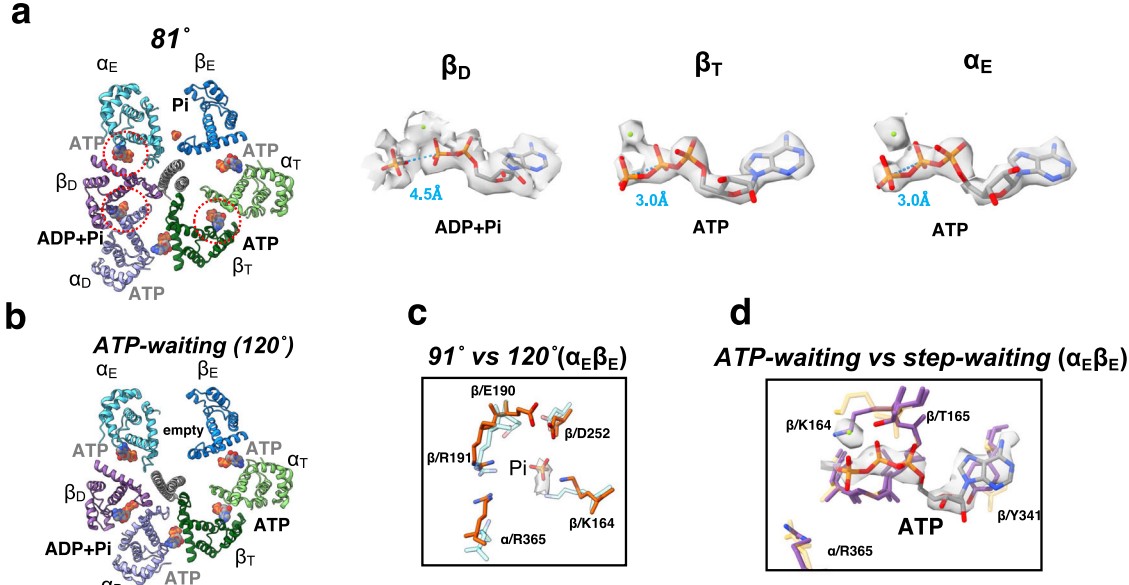

**Fig. 4 | Structures captured at low [ATP]. a** Cross section of the $F_1$ domain in 81° showing the catalytic sites viewed from the $F_o$ side. The bound nucleotides are represented as spheres. The electron density of the nucleotides within the enclosed circles are shown in larger view on the right. The distance between γ (or Pi) and β phosphate of the ATP in each conformational state is shown in blue (Å). Each distance was calculated using Chimera software. **b** Cross section of the $F_1$ domain in *ATP-waiting* (120°). **c** Comparison of the Pi binding site in $α_Eβ_E$ in the 91° and 120° structures with electron density indicated for the Pi. **d** Comparison of the catalytic site of the $α_Eβ_E$ in *ATP waiting* (cream) and *step waiting* (purple). ATP in *step waiting* is represented by the orange and blue sticks. The *step waiting* structure was captured at high [ATP].

## The 80° dwell

A further major insight is that the 80° dwell during the 120° step observed in $F_1$-ATPase is the result of waiting for ATP hydrolysis at the $α_Dβ_D$. In the $α_Dβ_D$ of *81°*, the density of ATP is confirmed, whereas ADP+Pi is bound in the *post-hyd* (83°) of $α_Dβ_D$ (Fig. 2c), indicating that hydrolysis of ATP is completed at $α_Dβ_D$ in *post-hyd*.

Notably, the $α_Dβ_D$ in 81° has the most closed conformation, adopting a slightly more open conformation in the following *post-hyd* (Fig. 2b). In the $α_Dβ_D$ of 81°, the γ-phosphate of ATP interacts with β/Glu190 and β/Arg192 residues, and the adenine ring interacts with the β/Tyr341 in the *CT* domain (Fig. 2c and Supplementary Fig. 5c). Therefore, ATP bound to the catalytic site of $α_Dβ_D$ stabilizes the closed conformation of the β subunit. Once ATP at the catalytic site of $α_Dβ_D$ is hydrolyzed, $α_Dβ_D$ is then able to structurally transition to a more open conformation (Supplementary Figs. 5c, d). Indeed, the structure of $α_Dβ_D$ gradually becomes more open as the rotation of the γ subunit proceeds, to the half open $α_Dβ_D^{HO}$ state visible in the 120° ATP/*step waiting* state (Fig. 3c). Based on the dwell time analysis at the 80° rotation angle of $F_1$, it has been proposed that two or more catalytic events, including ATP hydrolysis and Pi release, contribute to the 80° dwell[16,17]. Our results support that the dwell at this rotation angle is due mainly to ATP hydrolysis, but is also dependent on the structural transition from the 81° to the *post-hyd*.

## The subsequent 40° rotation step

The third finding is that the final 40° rotation of the 120° step is composed of three short sub-steps (Fig. 5), that are largely uncoupled from ATP hydrolysis occurring in the $F_1$ domain. In previously suggested rotational model of $F_1$, the dissociation of Pi at 80° is reported to drive the last 40° of rotation (Fig. 1d)[17,27]. However our structures reveal that the Pi remains both following rotation beyond 80°, and indeed the structure of $α_Eβ_E$ without Pi in 101° indicates that release of Pi occurs during the structural transition from 91° to 101°. The rotation of the γ-subunit associated with Pi release is only 10° (Fig. 5f, g), suggesting that the contribution of conformational changes to Pi release (or Pi release by conformational change) is small. Other conformational changes, such

as the 8° step of *post-hyd* to 91° and the 19° step between 101° and 120°, occur independently of the ATP hydrolysis cycle. In addition, the $α_Tβ_T$ of *ATP-waiting* adopts a more closed conformation after the structural transition from *120°* to *step-waiting* without rotation of the γ subunit (Fig. 3e). The only explanation for these spontaneous conformational rearrangements, uncoupled from chemical reactions, is that they are caused by the release of strain inside the molecule. The 80° and 120° (0°) structures identified in the $F_1$ domain of $F_oF_1$ are also observed in $F_1$-ATPase[22]. Therefore, the strain inside the molecule that drives the 40° step is most likely of $F_1$ origin. For instance, a comparison of the γ subunit in the 101° and 120° structures shows a slight structural difference (Supplementary Fig. 9). Together, our results strongly suggest that these structural changes, including the 10° step of *91°* associated with Pi release, is driven by the release of the internal molecular strain accumulated by the initial 80° rotation.

## Principle of rotation mechanism of ATP synthases

Taken together, these structural snapshots of the $F_1$ domain indicate that $F_oF_1$ functions via a similar mechanism to the ATP-driven rotation of V/A-ATPase that we have previously demonstrated[21] (Supplementary Fig. 10a). The binding of ATP to empty $AB_{open}$, corresponding to $α_Eβ_E$, in $V_{2nuc}$ with two nucleotides bound forms $V_{3nuc}$ with nucleotides bound to all three AB dimers ($AB_{open}$ with ATP, $AB_{semi}$ with ATP, and $AB_{closed}$ with ADP and Pi). Three distinguishable catalytic events occur at the three AB dimers simultaneously and these events contribute to the first 120° step of the rotor in a concerted manner (Supplementary Fig. 10b)[21]. Considering each 120° step, the rotation mechanism of $F_oF_1$ and V/A-ATPase is almost identical. ATP binds to the $F_1$ domain of *ATP-waiting*, resulting in the *step-waiting* where the three catalytic sites are occupied with nucleotides, corresponded to $V_{3nuc}$. *Step-waiting* comprises $α_Eβ_E$ with ATP, $α_Tβ_T$ with ATP, and $α_Dβ_D^{HO}$ with ADP and Pi. Each conformational change, from $α_Eβ_E$ to $α_Tβ_T$, $α_Tβ_T$ to $α_Dβ_D^{HO}$, and $α_Dβ_D^{HO}$ to $α_Eβ_E$ occurs simultaneously, accompanied by the full 120° rotation of the of γ-subunit, hydrolysis of ATP in the $α_Tβ_T$ and release of products (ADP and Pi) from $α_Dβ_D^{HO}$ (Supplementary Fig. 10d). This structural transition is driven in part by the change from the

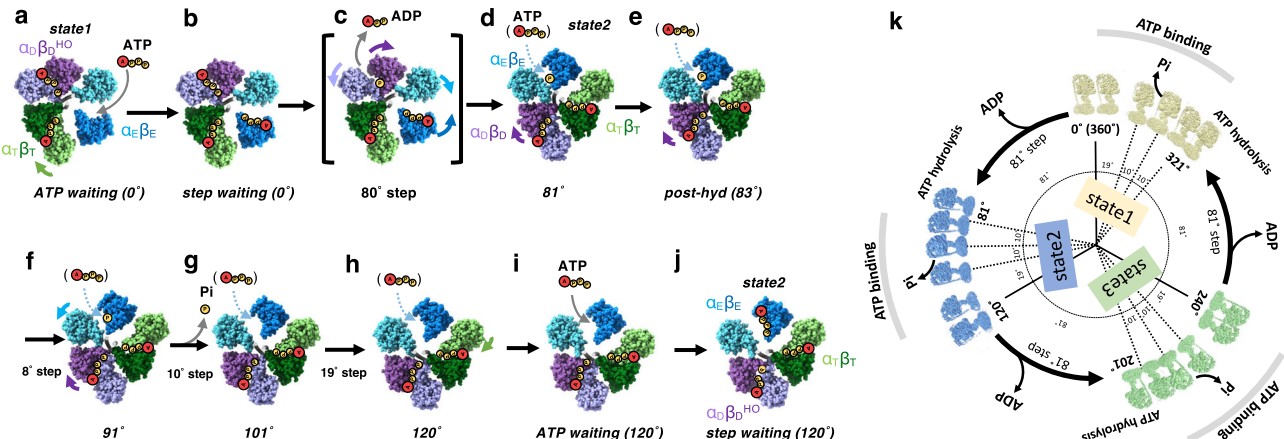

**Fig. 5 | ATP driven rotation scheme of F$_o$F$_1$. a** Under low [ATP] conditions, the catalytic site in α$_E$β$_E$ of *ATP waiting* remains empty. **b** The *step-waiting* is formed by binding of ATP to α$_E$β$_E$ of *ATP waiting*. **c** The *step-waiting* initiates the 80° rotation step of the γ subunit coupled with structure transition of the three αβ dimers; α$_E$β$_E$ to α$_T$β$_T$ with a zippering motion caused by binding of ATP to α$_E$β$_E$, α$_T$β$_T$ to α$_D$β$_D$, and α$_D$β$_D$$^{HO}$ to α$_E$β$_E$ with associated release of ADP via an unzipping motion of α$_D$β$_D$$^{HO}$. **d** ATP bound to α$_T$β$_T$ is hydrolyzed in the α$_D$β$_D$ dimer of 81° just after the 81° rotation. **e, f** Once ATP bound to α$_D$β$_D$ is hydrolyzed, an unzipping motion of α$_D$β$_D$ (*purple arrows*) proceeds via a 10° rotation step of γ, resulting in *the* structural change of *hydrolysable* to 91° through *post-hyd*. The outward motion of α$_E$ in 91°(*light blue arrow*) in concert with a further 10° rotation induces release of Pi,

resulting in *101°*which adopts a more open α$_E$β$_E$. **g** The final rotation from 101° to 120° occurs without structural change in any of three catalytic dimers. **h** Further motion of the *CT* domain of α$_E$ induces the structural between 120° and, **i** *ATP waiting* without any associated rotation of the γ subunit. **j** The *step-waiting* (120°) is formed by binding of ATP to α$_E$β$_E$ of *ATP waiting*. The α$_E$β$_E$ of six intermediates (81°, *post-hyd*, 91°, 101°, 120°, *and ATP waiting*) are occupied with ATP at high [ATP], indicating that ATP binds to empty α$_E$β$_E$ at any rotation angle of γ (*light blue dash arrows*). **k** 360° rotation of F$_o$F$_1$. CryoEM maps of F$_o$F$_1$ obtained at high [ATP] are placed in a circular arrangement according to the angle of the γ subunit. The three state maps are represented by yellow (state1), blue (state2), and green (state3), respectively.

ATP-bound α$_E$β$_E$ state to the more closed α$_T$β$_T$ state resulting from a zipper motion of α$_E$β$_E$ that occurs upon binding ATP[28]. Our structural analysis indicates that the ATP bound to α$_T$β$_T$ is hydrolyzed as a result of the conformational change from α$_T$β$_T$ to α$_D$β$_D$ which occurs just after the 80° rotation angle (81° to 83°) (Fig. 5). This subsequently leads to the conformational change of α$_T$β$_T$ to α$_D$β$_D$$^{HO}$ that occurs during the further rotation to 120° (Supplementary Fig. 10d). The conformational change from α$_T$β$_T$ to α$_D$β$_D$$^{HO}$ is likely to occur spontaneously since it involves ATP hydrolysis, an exergonic reaction. The full 120° rotation of the γ-subunit, coupled with spontaneous reactions occurring at both α$_E$β$_E$ with ATP and α$_T$β$_T$ with ATP, reduces the affinity for ADP and Pi at α$_D$β$_D$$^{HO}$, resulting in the release of ADP during the first 80° rotation step and Pi during the following 40° rotation step.

## Discussion

The proposed mechanism of rotation of F$_o$F$_1$ in this study differs considerably from the model proposed by previous cryo-EM structural snapshots of the F$_1$-ATPase, where the 40° rotation is driven by hydrolysis of ATP accompanied by simultaneous release of ADP and Pi[22]. This difference may be due to the use of a mutant F$_1$-ATPase (β/E190D) that significantly slows ATP hydrolysis at α$_D$β$_D$. As a result, only the 80° structure with α$_D$β$_D$ bound to ATP was identified in the previous study, which would have led to the conclusion that the 40° step was driven by ATP hydrolysis.

This study provides insight into the structural changes occurring in the F$_1$ domain during ATP hydrolysis. In the proton-driven ATP synthesis reaction, ADP is used as a substrate and is bound to the α$_E$β$_E$ instead of ATP. The structural changes observed during ATP hydrolysis are the reverse of those that occur during ATP synthesis. Moreover, our findings shed light on the reason why ATP synthases prefer ADP as a substrate. The obtained structure reveals that P$_i$ can bind to the α$_E$β$_E$ with or without nucleotide binding, and that this Pi binding can prevent the closing of the ATP-bound α$_E$β$_E$. As a result, ATP binding to α$_E$β$_E$ is inhibited, with ADP preferentially binding.

In addition, our study demonstrates that the F$_o$F$_1$ ATP synthase uses the chemical energy of ATP hydrolysis to drive the 80° rotation

of the γ subunit and this causes the internal molecular strain which drives the last 40° rotation step, resulting in minimal heat dissipation of the chemical energy. As a result, the coupling of the chemo-mechanical cycle in F$_1$ domain is achieved at almost 100 % efficiency, as previously demonstrated for F$_1$-ATPase by single molecule observation experiments[2,18,19].

Under high ATP conditions, the presence of ADP resulting from ATP hydrolysis suggests the potential existence of an ADP-inhibited structure in the reaction solution. Indeed, the addition of Lauryldimethylamine oxide (LDAO), which is known to enhance the activity of F$_1$-ATPase in the ADP-inhibited state[25], significantly increased the ATPase activity of F$_o$F$_1$ used in this experiment (Supplementary Fig. 1d). This suggests that ADP-inhibited structure of F$_o$F$_1$ is present in the reaction mixture. However, the structure identified as the ADP-inhibited state have not been obtained in this study. Previous studies have suggested that the dissociation of Pi, resulting from ATP hydrolysis, occurs before the formation of the ADP-inhibited structure[29]. However, in the structures obtained in this study, ADP was found to bind in the presence of Pi in all catalytic sites, indicating that these structures represent rotational intermediates rather than a dead-end structure, such as the ADP-inhibited structure.

One possible reason for the inability to isolate structures definitively identified as ADP-inhibited conformations is that the ADP-inhibited structures and the rotational state conformations closely resemble each other, making it challenging to classify them as distinct classes. The specific characteristics of the ADP-inhibited structure remain unclear. Further investigations using conditions that induce ADP inhibition or conducting snapshot analysis of mutant variants susceptible to ADP inhibition are necessary to identify the ADP-inhibited structure.

## Methods

### Sample preparation

For ΔεCT-F$_o$F$_1$ ATP synthase from *G. stearothermophilus*, we deleted the C-terminal domain (83–133 amino acids) of ε from the expression vector PTR19-ASDS[27]. The expression plasmid was transformed into *E. coli* strain DK8 in which the endogenous ATP synthase was deleted.

Transformed *E. coli* cells were cultured in 2xYT medium for 24 h. Cultured cells were collected by centrifugation at 5000 x g and suspended in lysis buffer (50 mM Tris-Cl pH 8.0, 5 mM $MgCl_2$, and 10% [w/v] glycerol). The cells were disrupted by sonication using Branson Sonifier. Membrane fraction was collected by centrifugation at 85,000xg for 30 min, then were resuspended in solubilization buffer (50 mM Tris-Cl, pH 8.0, 5 mM $MgCl_2$, 10% [w/v] glycerol, and 2% *n-dodecyl-D*-maltoside [DDM]). The suspension was mixed for 1 hr at room temperature, then ultracentrifuged at 85,000 x *g*. The enzyme in the supernatant was affinity purified using a nickel-nitrilotriacetic acid ($Ni^{2+}$-NTA) column. For protein samples used under high ATP conditions, endogenous nucleotides were removed from $\Delta \varepsilon CT$-$F_oF_1$ by dialysis with phosphate EDTA buffer containing 200 mM sodium phosphate, pH 8.0, 10 mM EDTA and 0.03% DDM at 25 °C for 24 h. Samples were concentrated to ~500 μl by an Amicon ultra (100 k cut), and then subjected to gel permeation chromatography using a Superose™ 6 Increase column equilibrated with 20 mM Tris-Cl, pH 8.0, 150 mM NaCl, 0.03 %DDM. Peak fractions containing $F_oF_1$ were collected and concentrated to ~300 μl using an Amicon ultra (100 k cut) for grid preparation.

### Measurement of ATPase activity
The ATPase activity of purified $F_oF_1$ was assessed at 25 °C using a NADH-coupled assay[30]. The assay mixtures contained 50 mM Tris-HCl (pH 8.0), 100 mM KCl, 5 mM $MgCl_2$, 2.5 mM phosphoenolpyruvate (PEP), 100 μg/ml pyruvate kinase (PK), 100 μg/ml lactate dehydrogenase, and 0.2 mM NADH, and a range of concentrations of ATP-Mg. The reaction was initiated by adding 2 pmol of $F_oF_1$ to 2 ml of the reaction mixture. ATPase activity of $F_oF_1$ was measured by monitoring NADH oxidation over time by absorbance at 340 nm.

### CryoEM grid preparation
Holey Quantifoil R1.2/1.3 Mo grids were used for high [ATP] conditions and UltraAuFoil R1.2 and 1.3 grid for low [ATP] conditions were used. Before using holly grids, they were treated to 1-min glow discharge by the Ion Bombarder (Vacuum Device).

For high [ATP] conditions, 3 μl of 200 mM ATP-Mg was added to 20 μl of 10 μM of enzyme solution, and then the solutions mixed well by pipetting. The mixtures containing 26 mM ATP-Mg, 9 μM enzyme, 17 mM Tris-Cl, pH 8.0, 130 mM NaCl, ~0.03% DDM were incubated for 20 sec at 25 °C. Then, 3 μL enzyme solution was loaded onto the grid and blotted for 6 sec with a blot force of 10, drain time of 0.5 sec, and 100% humidity using a FEI Vitrobot (*ThermoFisher*), followed by flash freezing by liquid ethane. For low [ATP] conditions, 2 μl of reaction buffer containing 0.2 M Tris-Cl pH 8.0, 250 μM ATP, 40 mM PEP, 1 M KCl, 5 mg/ml PK was added to 18 μl of 10 μM enzyme solution, and then mixed well by pipetting. The reaction mixtures containing enzyme were incubated at 25 °C for 60 sec. Aliquots of 3 μL of the reaction mixtures were loaded onto grids and immediately flash frozen as described above. The prepared cryo-grids were stored in liquid nitrogen until use.

### Cryo-EM data acquisition
Cryo-EM imaging was performed using a Titan Krios (FEI/Thermo Fisher Scientific) operating at 300 kV acceleration voltage and equipped with a K3 electron detector (Gatan) in electron counting mode (CDS). SerialEM software was used for data collection. CryoEM movies were collected at a nominal magnification of 88,000 with a pixel size of 0.88 Å/pix. The defocus range was 0.8-2.0 μm, and data were collected at 50 frames. The total electron dose was 60 electrons/$Å^2$.

### Image processing
The detailed procedures for single-particle analysis are summarized in Supplementary Figs. 2 and 3. RELION 4.0[31] and CryoSPARC v3.3[32] were used for image analysis. The conversion of file formats between

RELION and CryoSPARC was executed using csparc2star.py in pyem. For both high ATP (5922 movies) and low ATP (8200 movies) conditions, beam-induced drift was corrected using MotionCor2[33], and CTF estimation was performed using CTFFIND4.1[34]. Particle picking was performed by Topaz[35]. In each dataset, good particles were sorted by 2D classification after template picking from hundreds of images, and 2000-6000 particles were used to train the Topaz model. 1,118,093 high [ATP] particles were and 2,654,860 low [ATP] particles were picked by Topaz and then these were subjected to 2D classification by CryoSPARC, which further selects particles of good quality. Heterogeneous refinement was used to further select particles and classify into different rotation states. The total number of particles selected and classified was 241,668 at high [ATP] and 557,503 at low [ATP] conditions. All particles were combined and a focused 3D refinement was performed on the $F_1$ domain. The $F_1$ structures with different rotation angles were classified by 3D classification without alignment with masks for the $F_1$ domain. Both CTF refinement and Bayesian polishing were carried out multiple times. Finally, we obtained structures with multiple rotational states at 2.6-4.2 Å resolution, estimated by the gold standard Fourier shell correlation (FSC) = 0.143 criterion (Supplementary Figs. 2c and 3c).

### Model building and refinement
The atomic model was built using the previous cryo-EM structure of bacterial $F_oF_1$ PDB *6N2Y*. The $\beta_{DP}$ has a very different conformation in our structures, so $F_1$ excluding $\beta_{DP}$ was fitted as a rigid body and $\beta_{TP}$ was fitted to the $\beta_{DP}$ density. The initial model was manually modified using COOT and ISOLDE[36]. The manually modified model was refined by the *phenix.real_space_refinement* program[37]. Manual corrections and refinement were iterated until model parameters were improved. The final model was evaluated by MolProbity[38] and EMRinger[39]. The statistical data and data corrections for all structures have been summarized in Supplementary Table 2A–D.

### Reporting summary
Further information on research design is available in the Nature Portfolio Reporting Summary linked to this article.

### Data availability
The Cryo-EM density maps and models generated in this study have been deposited in the EM and protein database under accession codes: EMD-34748, EMD-34749, EMD-34750, EMD-34751, EMD-34752, EMD-34753, EMD-34754, EMD-34755, EMD-34756, EMD-34757, EMD-34758, EMD-34760, EMD-34759, EMD-34761, EMD-34763, EMD-34762, EMD-34764, EMD-34765, EMD-34766, EMD-34767, EMD-35373, EMD-34768, EMD-34769, EMD-34770, EMD-34771, EMD-34772, EMD-34773, EMD-34774, EMD-34775, EMD-34776, EMD-34777, EMD-34779, EMD-34780, EMD-34778, EMD-34781, EMD-34782, EMD-34783, EMD-34784, EMD-34785, EMD-34786, EMD-34787, EMD-34788, EMD-34789, and EMD-34790 and PDB 8HH1, 8HH2, 8HH3, 8HH4, 8HH5, 8HH6, 8HH7, 8HH8, 8HH9, 8HHA, 8HHB, and 8HHC. Previously published protein structures used in this study are also available from the PDB for 6N2Y. Source data are provided with this paper.

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

## Acknowledgements

We are grateful to all the members of the Yokoyama Lab for their continuous support and technical assistance. Our research was supported by Grant-in-Aid for Scientific Research (JSPS KAKENHI) Grant Numbers 20H03231 to K.Y., 20K06514 to J.K., and Takeda Science foundation funding to K.Y. Our research was also supported by the Platform Project for Supporting Drug Discovery and Life Science Research (Basis for Supporting Innovative Drug Discovery and Life Science Research (BINDS)) from AMED under Grant Number JP17am0101001 (support number 1312), and Grants-in-Aid from the "Nanotechnology Platform" of the Ministry of Education, Culture, Sports, Science and Technology (MEXT). This work was also supported by JST CREST to K.M. (Grant Number. JPMJCR1865).

## Author contributions

K.Y., A.N. designed, performed and analyzed the experiments. A.N. analyzed the data and contributed to preparation of the samples. J.K. and K.M. provided technical support and conceptual advice. K.Y. designed and supervised the experiments and wrote the manuscript. All authors discussed the results and commented on the manuscript. All data is available in the manuscript or in the supplementary materials.

## Competing interests

The authors declare no competing interests.
