## [Peer Review File · Nature Communications]

REVIEWER COMMENTS

Reviewer #1 (Remarks to the Author):

The manuscript "Mechanism of ATP hydrolysis dependent rotation of ATP synthases" by Yokoyama et al. reports on the results of a cryo-electron microscopy study of the thermophilic bacterial F-type ATP synthase from *Geobacillus stearothermophilus*. This bacterium is a model organism for ATP synthase research and is commonly known as *Bacillus PS3* sp. The authors expressed the enzyme heterologously in *Escherichia coli*, purified it by Ni-NTA chromatography, and examined two types of samples: nucleotide-free ATP synthase with ATP added to 26 mM and with ATP added to 25 μ M. The enzyme was mixed with the nucleotide, and after a brief moment of time (tens of seconds), it was instantly frozen and examined in Cryo-EM. The data collected (\sim 240k particles for high ATP and \sim 560k particles for low ATP) were analyzed, and a series of structures (with resolution from 2.6 to 4.2 Å) were obtained for both samples.

In the high ATP sample, the authors observed six structural intermediates, and in the low ATP sample, they observed five intermediates. It was possible to determine the occupancies of all the catalytic sites (nucleotides and phosphate), and the authors suggested a scheme of catalytic events based on this information. An important new finding is that phosphate release seems to occur not at the 80° but at 91°, i.e. the rotation from the 80° position is not driven by phosphate release as previously considered.

The results reported in the manuscript are sound and provide important new insights into the mechanism by which rotary ATP hydrolysis proceeds. The study's findings shed light on the molecular basis of ATP-driven rotation in ATP synthases and could have important implications for our understanding of energy production and consumption in living systems and of protein nanomachines functioning. There are some issues in the text that need clarification before publication, namely:

1) It is unclear why the authors used nucleotide-depleted enzyme for high ATP experiments, while for low ATP experiments, nucleotide depletion was omitted. It is also not clear how the efficiency of the depletion was verified. Were all nucleotides depleted, including those in non-catalytic sites? Experiments that directly confirm the amount of bound nucleotides should be performed for both ND-FoF1 and FoF1 without nucleotide depletion.

2) Original traces of the ATPase activity measurements for ND-FoF1 and non-ND-FoF1 should also be included in the manuscript (in the supplementary data). It is well-known that in the ATP-regenerating system, nucleotide-depleted F1 of *G. stearothermophilus* hydrolyzes ATP in three kinetic phases. The authors should clarify whether they observed similar kinetics and how long the initial stage of high-rate hydrolysis was, as the ND-FoF1 sample used for high-ATP structure determination was taken

approximately 25-30 seconds after the addition of ATP. According to the cited Paik et al. (1994) <https://pubmed.ncbi.nlm.nih.gov/8286329/>, 30 seconds after reaction start, the enzyme is already lapsing into an ADP-inhibited state. This is important to correctly interpret the structural data obtained by cryo-EM for the high ATP samples.

3) The authors should explain in the discussion whether they observed a fraction of FoF1 in an ADP-inhibited state. If not, they should explain why not. If they did observe such a fraction, they should report the proportion of ADP-inhibited FoF1 molecules to active FoF1.

4) 26 mM of ATP is well above any physiological ATP concentration (cellular ATP concentration does not exceed 10 mM). It should be explained in the manuscript why such a high concentration was chosen. (In Extended Data Figures 2 and 3, it is stated in the figure captions that ATP concentrations were 5 mM and 50 μ M, respectively. Why is it different from the 26 mM and 25 μ M in the main text of the manuscript? Have the authors obtained structures with 5 mM and 50 μ M ATP as well?) The concentration of magnesium is also unclear. Was it also 26 mM Mg in the case of 26 mM ATP? It is stated in line 359 of the manuscript that 3 μ l of 200 mM ATP-Mg was added to the enzyme solution. Was it actually a stock solution with 200 mM ATP and 200 mM Mg? Please provide detailed information about the stock solution (concentration of salts, pH) and the final composition of the buffer where the enzyme was frozen for Cryo-EM.

Minor points:

i) It should be stated in the abstract that the experiments were done on *Bacillus PS3* sp. enzyme (I recommend explicitly stating that *Geobacillus stearothermophilus* was previously known as *Bacillus PS3* since a lot of ATP synthase literature uses the latter name). I also recommend changing the title to "Mechanism of ATP hydrolysis-dependent rotation of bacterial ATP synthase" since enzymes from mitochondria might have substantial deviations from the pattern found by the authors for *G. stearothermophilus* FoF1.

ii) Line 33: It might well be that the majority of the ATP is actually synthesized by photophosphorylation, not by OxPhos, so it is probably better to mention both.

iii) Line 41: Typo, Fo should be used instead of F0.

iv) Line 75: Please specify the organism from which the V/A ATPase was characterized.

v) Lines 333-334: How was the membrane fraction obtained? What was the cell disruption procedure? Please provide a more detailed description.

After the issues above are addressed, I think the manuscript can be published in Nature Communications. I have no doubts that it will be of great interest to a broad spectrum of researchers, including those from the fields of biochemistry, protein biophysics, bioenergetics, and microbiology."

Reviewer #2 (Remarks to the Author):

Please refer to the attached document to see Reviewer 2's comments in full.

The work is quite significant to the field. However, due the resolution, some data interpretation is questionable. In addition, the manuscript is difficult to follow. Significant improvement of the manuscript is needed before being considered for publication.

Reviewer #3 (Remarks to the Author):

This is a very impressive study that uses cryo-EM to identify the reaction mechanism of ATP hydrolysis. The study uses a bacterial enzyme that has been modified to allow ATP hydrolysis without auto-inhibition. The team uses 2 ATP:Mg concentrations to trap different reaction intermediates. The results are impressive where they are able to identify multiple new states with bound substrate and products. The details are likely of most interest to the scientists in the field, while that movie will be interest to the general audience.

There is one state where the authors find Pi bound to the same site as ATP. The Pi binding site is identified, but it might be helpful to relate the Pi binding site that observed in the crystal structure of yeast F1. V. Kabaleeswaran, et al, EMBO J 2006: 25, 5433-5442.

It is also curious why the authors 1. stripped the enzyme of nucleotides, 2. added such high Mg:ATP, and 3. why the reaction was allowed to go so long, 20 min, after addition of ATP.

Yokoyama and colleagues presented the cryo-EM structures of a bacterial F_0F_1 ATP synthase in multiple intermediate states. By capturing structures of the F1 domain in 6 intermediate states in the presence of high ATP or low ATP, the authors have provided in-depth understanding of the chemo-mechanical coupling mechanism by F_0F_1 ATP synthase. The structural analyses are extensive, but the manuscript can certainly be improved.

1. The wild type F_0F_1 ATP synthase has low activity in ATP hydrolysis. In contrast, the truncated version used in this study displays high activity in hydrolyzing ATP. The structures revealed that ATP binding, hydrolysis, and P_i release can trigger sub-step conformational changes of the F1 domain. Under physiological conditions, the F_0F_1 ATP synthase functions to bind ADP and P_i to synthesize ATP. What is the implication of this study to our understanding of ATP synthesis by the F_0F_1 ATP synthase? The author should discuss this in the manuscript.
2. In Figure 2c, the authors assigned densities as P_i in the post-hyd and hydrolysable states. However, the densities are continuous with the ATP molecules, how confident it is to assign the densities as P_i and how the author assign the specific positions of P_i in continuous densities?
3. In Fig. 2d, I had similar concerns as raised in Question 2. To me, the P_i groups can also be assigned as Magnesium ions, which seems to fit better to the chemical environment.
4. It is very difficult to tell the ATP binding states from Figure 2a and 2b. I would suggest labelling the nucleotides in these panels.
5. Figures 3 & 4, the conformational differences in panels a, b, and c were poorly displayed. It is extremely difficult for readers to tell.
6. There are a few Figure mis citations. For examples, Line 120, Fig. 3c should be Fig. 2c. Furthermore, many figures and extended figures were cited in a messed order. It is very difficult to follow by readers.

Our responses to all reviewers' comments

Reviewer #1

reviewer comments

The manuscript "Mechanism of ATP hydrolysis dependent rotation of ATP synthases" by Yokoyama et al. reports on the results of a cryo-electron microscopy study of the thermophilic bacterial F-type ATP synthase from *Geobacillus stearothermophilus*. This bacterium is a model organism for ATP synthase research and is commonly known as *Bacillus PS3* sp. The authors expressed the enzyme heterologously in *Escherichia coli*, purified it by Ni-NTA chromatography, and examined two types of samples: nucleotide-free ATP synthase with ATP added to 26 mM and with ATP added to 25 μ M. The enzyme was mixed with the nucleotide, and after a brief moment of time (tens of seconds), it was instantly frozen and examined in Cryo-EM. The data collected (~240k particles for high ATP and ~560k particles for low ATP) were analyzed, and a series of structures (with resolution from 2.6 to 4.2 Å) were obtained for both samples.

In the high ATP sample, the authors observed six structural intermediates, and in the low ATP sample, they observed five intermediates. It was possible to determine the occupancies of all the catalytic sites (nucleotides and phosphate), and the authors suggested a scheme of catalytic events based on this information. An important new finding is that phosphate release seems to occur not at the 80° but at 91°, i.e. the rotation from the 80° position is not driven by phosphate release as previously considered.

The results reported in the manuscript are sound and provide important new insights into the mechanism by which rotary ATP hydrolysis proceeds. The study's findings shed light on the molecular basis of ATP-driven rotation in ATP synthases and could have important implications for our understanding of energy production and consumption in living systems and of protein nanomachines functioning. There are some issues in the text that need clarification before publication.

our response

We thank the reviewer for their careful analysis of our research and for the very POSITIVE comments.

1-1 reviewer comment

1) It is unclear why the authors used nucleotide-depleted enzyme for high ATP experiments,

while for low ATP experiments, nucleotide depletion was omitted. It is also not clear how the efficiency of the depletion was verified. Were all nucleotides depleted, including those in non-catalytic sites? Experiments that directly confirm the amount of bound nucleotides should be performed for both ND-FoF1 and FoF1 without nucleotide depletion.

I-1 our response

To exclude the possibility that the nucleotides present in the resulting structure were endogenous, nucleotide-depleted (ND) FoF1 was used in the experiments. Nucleotide depletion using this method was recently reported (Nakano et al. 2022, *PNAS Nexus*). In the low [ATP] reaction condition, the reaction solution contains 9 μM FoF1 and 25 μM ATP; using ND-FoF1 under these conditions is likely to deplete the ATP in the reaction solution as the majority of the ATP binds to the α subunit. Therefore, FoF1 without nucleotide depleted treatment was used at low [ATP]. The binding of ATP to all three α subunits of FoF1 without ND-treatment has been demonstrated in previous structural papers (Nakano et al. 2022, *PNAS Nexus*, Guo et al, 2019 *Elife* 2019). To emphasize this point, the manuscript was revised as follows;

p4 Line 110;

To exclude the possibility that the nucleotides present in the resulting structure were endogenous, we used ND-FoF1 without endogenous nucleotides for structural analysis at high [ATP]. When conducting reactions at low [ATP], the reaction solution typically contains 9 μM FoF1 and 25 μM ATP. Therefore, using ND-FoF1 at low [ATP] is likely to lead to ATP depletion in the solution, as most of the ATP will bind to the three α subunits in F1 domain. To avoid this issue, we used FoF1 without nucleotide depletion treatment for structural analysis at low [ATP].

I-2 reviewer comments

2) Original traces of the ATPase activity measurements for ND-FoF1 and non-ND-FoF1 should also be included in the manuscript (in the supplementary data). It is well-known that in the ATP-regenerating system, nucleotide-depleted F1 of *G. stearothermophilus* hydrolyzes ATP in three kinetic phases. The authors should clarify whether they observed similar kinetics and how long the initial stage of high-rate hydrolysis was, as the ND-FoF1 sample used for high-ATP structure determination was taken approximately 25-30 seconds after the addition of ATP. According to the cited Paik et al. (1994) <https://pubmed.ncbi.nlm.nih.gov/8286329/>, 30 seconds after reaction start, the enzyme is already lapsing into an ADP-inhibited state.

This is important to correctly interpret the structural data obtained by cryo-EM for the high ATP samples.

1-2 our response

As instructed by the reviewer, the ATP hydrolysis profile has been added as supplementary figure S1. Unlike the results obtained by Paik et al. using F₁-ATPase, we did not see an initial lag phase but rather observed a gradual increase in ATPase activity in the purified F_oF₁ used in our study. This suggests that F_oF₁ transitions from the initially inactive conformation to the active conformation and does not fall into the ADP inhibitory conformation; however, at present we do not have a satisfactory explanation for the difference in our results with those of the Paik et al study.

1-3 reviewer comment

3) The authors should explain in the discussion whether they observed a fraction of F_oF₁ in an ADP-inhibited state. If not, they should explain why not. If they did observe such a fraction, they should report the proportion of ADP-inhibited F_oF₁ molecules to active F_oF₁.

1-3 our response

As mentioned in our response to the previous comment, we analyzed the structure of F_oF₁ undergoing the transition to the active form. This suggests that there is no ADP-inhibited form in the reaction mixture. Analyzing the structure of F_oF₁ solution inhibited by ADP is clearly an important next step and something we are aiming to obtain soon. In order to clarify this point, we added the following to the text as suggested by the reviewer;

p3 Line 102

In contrast to the results obtained by Paik et al. using F₁-ATPase, our purified F_oF₁ showed no initial lag phase (Extended Data Fig. 1c). Instead, the ATPase activity of the enzyme gradually increased, indicating that F_oF₁ transitions from an initial inactive state to an active state without adopting the ADP inhibited state.

1-4 reviewer comment

4)-1 26 mM of ATP is well above any physiological ATP concentration (cellular ATP concentration does not exceed 10 mM). It should be explained in the manuscript why such a high concentration was chosen.

1-4 our response

The reaction mixture initially contained 9 μM of F_0F_1 , and assuming an average turnover rate of 100 molecules of ATP per second, the expected consumption of ATP in a 20-second reaction would be 18 mM. To avoid depletion of ATP, the initial concentration of ATP in the reaction mixture was set to 26 mM. However, we observed a gradual increase in the activity of F_0F_1 over the course of the 20-second reaction, suggesting that the actual consumption of ATP in the reaction mixture may be less than the predicted value. To highlight this point, we have added the following description of this issue.

p4 Line 121

To prevent depletion of ATP in the reaction mixture due to the ATPase activity of F_0F_1 , the concentration of ATP in the reaction mixture was set to 26 mM. This ensured that even after a 20-second reaction time, a saturating concentration of ATP remained in the solution, allowing us to obtain the structure under ATP-saturated conditions.

1-5 reviewer comments

4)-2 (In Extended Data Figures 2 and 3, it is stated in the figure captions that ATP concentrations were 5 mM and 50 μM , respectively. Why is it different from the 26 mM and 25 μM in the main text of the manuscript? Have the authors obtained structures with 5 mM and 50 μM ATP as well?).

1-5 our response

Thank you for the reviewer's feedback. The description in the extended figure was incorrect, and the ATP concentration is the same as described in the main text. We have corrected the legends.

1-6 reviewer comments

4)-3 The concentration of magnesium is also unclear. Was it also 26 mM Mg in the case of 26 mM ATP? It is stated in line 359 of the manuscript that 3 μl of 200 mM ATP-Mg was added to the enzyme solution. Was it actually a stock solution with 200 mM ATP and 200 mM Mg? Please provide detailed information about the stock solution (concentration of salts, pH) and the final composition of the buffer where the enzyme was frozen for Cryo-EM.

1-6 our response

In order to prepare the ATP-Mg solution used in our experiment, we added an equimolar amount of MgCl_2 to ATP solution. As a result, the reaction mixture contained the same molar amount of Mg ions as ATP. This information has been described in detail in the Methods

section of the manuscript. Additionally, following the reviewer's suggestion, we have included the final concentration of the reaction mixture in the Methods section on page 11. Please refer to the updated manuscript for more details.

p11 Line 385

For high [ATP] conditions, 3 μ l of 200 mM ATP-Mg was added to 20 μ l of 10 μ M of enzyme solution, and then the solutions mixed well by pipetting. **The mixtures containing 26 mM ATP-Mg, 9 μ M enzyme, 17 mM Tris-Cl, pH 8.0, 130 mM NaCl, ~0.03% DDM were incubated for 20 sec at 25 °C.**

1-7 reviewer comments

Minor points:

i) It should be stated in the abstract that the experiments were done on *Bacillus PS3 sp.* enzyme (I recommend explicitly stating that *Geobacillus stearothermophilus* was previously known as *Bacillus PS3* since a lot of ATP synthase literature uses the latter name). I also recommend changing the title to "Mechanism of ATP hydrolysis-dependent rotation of bacterial ATP synthase" since enzymes from mitochondria might have substantial deviations from the pattern found by the authors for *G. stearothermophilus* FoF1.

1-7 our response

Thank you for the reviewer's insightful suggestion. Following the suggestion, we have modified the Title and abstract as follows:

Title : **Mechanism of ATP hydrolysis dependent rotation of bacterial ATP synthase**

Abstract section

Here we describe catalytic intermediates of the F₁ domain **in F_oF₁ synthase from *Bacillus PS3 sp.*** during ATP mediated rotation captured using cryo-EM.

1-8 reviewer comments

ii) Line 33: It might well be that the majority of the ATP is actually synthesized by photophosphorylation, not by OxPhos, so it is probably better to mention both.

1-8 our response

We have modified the introduction section as suggested:

The majority of ATP, the energy currency of life, is synthesized by oxidative phosphorylation or **photophosphorylation**, catalysed by ATP synthase.

1-9 reviewer comment

iii) Line 41: Typo, Fo should be used instead of F0.

iv) Line 75: Please specify the organism from which the V/A ATPase was characterized.

1-10 our response

We have corrected these points in the text.

p3 Line 76

Previously, we have determined several intermediate structures of V/A-ATPase from *Thermus thermophilus* revealing an unprecedented level of detail for the ATP hydrolysis cycle in these enzymes.

1-11 reviewer comment

v) Lines 333-334: How was the membrane fraction obtained? What was the cell disruption procedure? Please provide a more detailed description.

1-11 our response

We added the requested details as follows:

p10 Line 356

Cultured cells were collected by centrifugation at 5000 x g and suspended in lysis buffer (50 mM Tris-Cl pH 8.0, 5 mM MgCl₂, and 10% [w/v] glycerol). **The cells were disrupted by sonication using Branson Sonifier. Membrane fraction was collected by centrifugation at 85,000xg for 30 min, then resuspended in solubilization buffer (50 mM Tris-Cl, pH 8.0, 5 mM MgCl₂, 10% [w/v] glycerol, and 2% *n*-dodecyl-*D*-maltoside [DDM]).** The suspension was mixed for 1 hr at room temperature, then ultracentrifuged at 85,000 x g.

1-12 reviewer comments

After the issues above are addressed, I think the manuscript can be published in Nature Communications. I have no doubts that it will be of great interest to a broad spectrum of researchers, including those from the fields of biochemistry, protein biophysics, bioenergetics, and microbiology."

1-12 our response

We thank the reviewer for their positive comments. We believe that we have addressed the reviewers comments in our manuscript.

Comments from Reviewer #2

2-1 reviewer comments

The work is quite significant to the field. However, due the resolution, some data interpretation is questionable. In addition, the manuscript is difficult to follow. Significant improvement of the manuscript is needed before being considered for publication.

2-1 our response

We agree with the reviewer that the work is significant in the field.

2-2 reviewer comments

1. The wild type FoF1 ATP synthase has low activity in ATP hydrolysis. In contrast, the truncated version used in this study displays high activity in hydrolyzing ATP. The structures revealed that ATP binding, hydrolysis, and Pi release can trigger sub-step conformational changes of the F1 domain. Under physiological conditions, the FoF1 ATP synthase functions to bind ADP and Pi to synthesize ATP. What is the implication of this study to our understanding of ATP synthesis by the FoF1 ATP synthase? The author should discuss this in the manuscript.

2-2 our response

The structural and functional analysis of F1-ATPase has been conducted with the understanding that 1. ATP synthesis and hydrolysis reactions in the F1 domain are reversible, and 2. understanding the ATP hydrolysis reaction in the F1 domain is key to understanding the reverse reaction of ATP synthesis. We share this perspective. Under physiological conditions, ADP, but not ATP, binds to the $\alpha_E\beta_E$ as a substrate, and ATP is synthesized from the bound ADP and Pi by conformational changes that are the reverse of those involved in the ATP hydrolysis reaction. In addition, our results provide important insight into why ATP synthases prefer ADP as a substrate. The structure obtained in the present study shows that Pi can bind to the $\alpha_E\beta_E$ with or without nucleotide binding. This Pi interferes with the closed movement of the ATP-bound $\alpha_E\beta_E$, thereby preventing the binding of ATP to the $\alpha_E\beta_E$ and causing preferential binding of ADP as a substrate.

We have added this point to the Discussion section as follows;

Discussion section

p10 Line 335

This study provides new insights into the structural changes occurring in the F_1 domain during ATP hydrolysis. In the proton-driven ATP synthesis reaction, ADP is used as a substrate and is bound to the $\alpha_E\beta_E$ instead of ATP. The structural changes observed during ATP hydrolysis are the reverse of those that occur during ATP synthesis. Moreover, our findings shed light on the reason why ATP synthases prefer ADP as a substrate. The obtained structure reveals that P_i can bind to the $\alpha_E\beta_E$ with or without nucleotide binding, and that this P_i binding can prevent the closing of the ATP-bound $\alpha_E\beta_E$. As a result, ATP binding to $\alpha_E\beta_E$ is inhibited, with ADP preferentially binding.

2-3 reviewer comments

In Figure 2c, the authors assigned densities as P_i in the post-hyd and hydrolysable states. However, the densities are continuous with the ATP molecules, how confident it is to assign the densities as P_i and how the author assign the specific positions of P_i in continuous densities?

2-3 our response

To confirm whether the density observed is a part of the ATP molecule or a released P_i , we evaluated the distance between the density and the β -phosphate group density, and the distance between the density and the interacting amino acid residues. Our analysis suggests that the assignment of P_i in the 81° and 83° structures under high ATP conditions in Figure 2c is valid, as demonstrated by the S.I Figure for reviewers. However, the density of P_i in the 81° structure obtained under low ATP conditions is less clear than in the 81° and 83° structures obtained under high ATP conditions. This is a valid concern raised by the reviewers. We have revised the manuscript to reflect that ATP hydrolysis occurs at some point during the conformational change from the 81° to the 83° structure. Additionally, we have changed the name of the 81° structure from "hydrolysable" to simply " 81° structure".

The revised sections are shown below.

p4 Line 139

The nucleotide bound to the $\alpha_D\beta_D$ differed between the two structures. In the $\alpha_D\beta_D$ of the 81° structure, ATP was identified at the catalytic site (Fig. 2c, left) and we termed the

structure as 81° . In contrast, the 83° structure obtained at high [ATP] contained ADP and Pi and is designated post-hydrolyzed (*post-hyd*) (Fig. 2c, *center*). This indicates that ATP bound to $\alpha_D\beta_D$ is hydrolyzed at 81° or between 81° and 83° , and that the structural change due to ATP hydrolysis is small.

p6 Line 211

The density of nucleotides in the $\alpha_D\beta_D$ at 81° under low [ATP] is not well defined (Fig. 4a, *right*). This implies that ATP and (ADP + Pi) are in equilibrium at the catalytic site of $\alpha_D\beta_D$ in 81° . Additionally, this observation suggests that ATP hydrolysis at the catalytic site does not directly cause rotation of the γ subunit, consistent with the binding change mechanism where ATP synthesis at the catalytic site is independent of rotation of the γ subunit¹.

p8 Line 259

A further major insight is that the 80° dwell during the 120° step, observed in F₁-ATPase, is the result of waiting for ATP hydrolysis at the $\alpha_D\beta_D$. In the $\alpha_D\beta_D$ of 81° , the density of ATP is confirmed, whereas ADP+Pi is bound in the 83° of $\alpha_D\beta_D$ (Fig. 2c), indicating that hydrolysis of ATP is complete at $\alpha_D\beta_D$ in 83° .

2-4 reviewer comment

In Fig. 2d, I had similar concerns as raised in Question 2. To me, the Pi groups can also be assigned as Magnesium ions, which seems to fit better to the chemical environment.

2-4 our response

A magnesium ion at the catalytic site is interacting with the amino group of β /T165 (Fig. 2d) and can be distinguished from Pi. On the other hand, Pi interacts with β /E190, β /R256, and β /R191 (Fig.2c and Extended Data Fig.6f), which is consistent with other crystal structures of F₁. Therefore, the assignment of Mg ions in this study is also reasonable.

2-5 reviewer comment

It is very difficult to tell the ATP binding states from Figure 2a and 2b. I would suggest labelling the nucleotides in these panels.

2-5 our response

Thank you for your comment. We have added nucleotide labels to Fig. 2a and 2b.

2-6 reviewer comments

Figures 3 & 4, the conformational differences in panels a, b, and c were poorly displayed. It is extremely difficult for readers to tell.

2-6 our response

We made significant improvements to the figure to highlight the differences between the structures. Specifically, in Figure 3, we enlarged the compared sections to emphasize the structural differences. Additionally, we have included a comparative figure of the structure of low ATP (Extended Data Figure 8). As a result of these changes, we believe that the differences between the structures are now more easily discernible.

2-7 reviewer comments

There are a few Figure mis citations. For examples, Line 120, Fig. 3c should be Fig. 2c. Furthermore, many figures and extended figures were cited in a messed order. It is very difficult to follow by readers.

2-7 our response

Thank you for your feedback. We have corrected all the incorrect parts and marked the changes in red.

Reviewer #3 (Remarks to the Author):

3-1 reviewer comments

This is a very impressive study that uses cryo-EM to identify the reaction mechanism of ATP hydrolysis. The study uses a bacterial enzyme that has been modified to allow ATP hydrolysis without auto-inhibition. The team uses 2 ATP:Mg concentrations to trap different reaction intermediates. The results are impressive where they are able to identify multiple new states with bound substrate and products. The details are likely of most interest to the scientists in the field, while that movie will be interest to the general audience.

3-1 our response

Thank you for your positive comment.

3-2 reviewer comment

There is one state where the authors find Pi bound to the same site as ATP. The Pi binding site is identified, but it might be helpful to relate the Pi binding site that observed in the crystal

structure of yeast F1. V. Kabaleeswaran, et al, EMBO J 2006: 25, 5433-5442.

3-2 our response

Thank you for the important comment. The Pi bound to the E site is in the same position as the Pi bound to the E site of yeast F1, as the reviewer pointed out. We have added this information to the manuscript, as follows;

p5 Line 149

The Pi binding site structure of $\alpha\epsilon\beta_E$ is very similar to that of the yeast F1 β_E Pi-binding site²⁶.

3-3 reviewer comment

It is also curious why the authors 1. stripped the enzyme of nucleotides, 2. added such high Mg:ATP, and 3. why the reaction was allowed to go so long, 20 min, after addition of ATP.

3-4 our response

Please refer to our response to Reviewer 1's comment 1-2 regarding this matter. The reaction time is not 20 minutes, but 20 seconds, and it is not a particularly long reaction time.

REVIEWERS' COMMENTS

Reviewer #1 (Remarks to the Author):

In the revised version of the manuscript, the authors have successfully addressed most of the issues I mentioned in the first review.

However, one point that is difficult for me to believe is that there was no ADP-inhibited form of the enzyme in the reaction mixture. Significant ADP inhibition is observed for Bacillus PS3 FoF1 even in an ATP-regenerating system, where virtually no ADP is present in the medium. Even in such conditions there is ADP-inhibition because of ADP resulting from ATP hydrolysis in the catalytic sites. In the experiments reported in this manuscript, at an ATP concentration of 26 mM, a significant (several mM?) amount of ADP must have been present in the medium after a few seconds, and it is unclear how the enzyme could possibly avoid ADP inhibition in such a situation.

The authors can easily check the extent of ADP inhibition directly by measuring the ATPase activity at 26 mM ATP with and without LDAO, a detergent that relieves ADP inhibition. I would be extremely surprised if there were no significant (several-fold) activation in the presence of LDAO. Since the authors used a truncated epsilon mutant, the activation by LDAO will reflect ADP inhibition only. I suggest that the authors either perform the experiment with LDAO and directly confirm the lack of an ADP-inhibited form in their experiment or make a careful update of the discussion section and provide some possible explanations for this situation.

Minor point: In the legend to Extended Data Figure 1 (c), please provide the composition of the measuring medium and the concentration of the enzyme (without the latter, the reader cannot estimate the activity).

Line 103: Make a lower index for "1" in F1.

After these issues are addressed, I believe the manuscript can be published.

Reviewer #2 (Remarks to the Author):

All my questions have been addressed. I am fine with the manuscript for being considered publication.

Reviewer #1

reviewer comments

In the revised version of the manuscript, the authors have successfully addressed most of the issues I mentioned in the first review.

However, one point that is difficult for me to believe is that there was no ADP-inhibited form of the enzyme in the reaction mixture. Significant ADP inhibition is observed for Bacillus PS3 FoF1 even in an ATP-regenerating system, where virtually no ADP is present in the medium. Even in such conditions there is ADP-inhibition because of ADP resulting from ATP hydrolysis in the catalytic sites. In the experiments reported in this manuscript, at an ATP concentration of 26 mM, a significant (several mM?) amount of ADP must have been present in the medium after a few seconds, and it is unclear how the enzyme could possibly avoid ADP inhibition in such a situation.

The authors can easily check the extent of ADP inhibition directly by measuring the ATPase activity at 26 mM ATP with and without LDAO, a detergent that relieves ADP inhibition. I would be extremely surprised if there were no significant (several-fold) activation in the presence of LDAO. Since the authors used a truncated epsilon mutant, the activation by LDAO will reflect ADP inhibition only. I suggest that the authors either perform the experiment with LDAO and directly confirm the lack of an ADP-inhibited form in their experiment or make a careful update of the discussion section and provide some possible explanations for this situation.

our response

We appreciate the insightful comments provided by Reviewer 1. In our study, we did not observe the transition to the ADP-inhibited state that has been observed in F1-ATPase. However, we observe an activation of FoF1 in the presence of LDAO, as shown in the referenced Figure. This suggests the possible presence of inactivated molecules. Nevertheless, we were unable to obtain a definitive structure that can be confidently identified as ADP inhibition, primarily due to the lack of a clear understanding of the specific characteristics of the ADP-inhibited structure.

In response to Reviewer 1's comments, we have incorporated an interpretation considering the potential presence of an ADP-inhibited structure at the end of the Discussion section in the revised manuscript.

p10, line349

Under high ATP conditions, the presence of ADP resulting from ATP hydrolysis suggests

the potential existence of an ADP-inhibited structure in the reaction solution. Previous studies have suggested that the dissociation of Pi, resulting from ATP hydrolysis, occurs prior to the formation of ADP-inhibited structure. However, in the structures obtained in this study, ADP was not found to exclusively bind without the presence of Pi in all catalytic sites, indicating that these structures were not ADP-inhibited structure. Nevertheless, the specific characteristics of the ADP-inhibited structure remain unclear. To identify the ADP-inhibited structure, it is necessary to conduct further investigations using conditions that induce ADP inhibition or perform snapshot analysis of mutant variants that are susceptible to ADP inhibition.

We have addressed the concerns raised by the reviewer1 and are confident that this manuscript is suitable for publication in *Nature Communications*.

Sincerely yours,

Ken Yokoyama

Referenced Figure

ATP hydrolysis profiles of $\Delta\epsilon CT-F_0F_1$ in the absence (blue) and presence of 0.1% LDAO (orange). The ATPase activity of $\Delta\epsilon CT-F_0F_1$ increased approximately twofold in the presence of 0.1% LDAO.

Twitter address

Lab address: @YOKOKEN15

Department address: @KSUSeimei

Hashtags: #ATP synthase, #FoF1, #CryoEM, #Rotary motor, #ATP